# Whole-Genome Sequencing Analyses Reveal the Evolution Mechanisms of Typical Biological Features of *Decapterus maruadsi*

**DOI:** 10.3390/ani14081202

**Published:** 2024-04-17

**Authors:** Wen-Jian Deng, Qian-Qian Li, Hao-Nan Shuai, Ren-Xie Wu, Su-Fang Niu, Qing-Hua Wang, Ben-Ben Miao

**Affiliations:** College of Fisheries, Guangdong Ocean University, Zhanjiang 524088, China; a653180374@163.com (W.-J.D.); lqq10142023@163.com (Q.-Q.L.); 15070559233@163.com (H.-N.S.); wurenxie@163.com (R.-X.W.); wqh31016877430@163.com (Q.-H.W.); miaobenben@stu.xmu.edu.cn (B.-B.M.)

**Keywords:** *Decapterus maruadsi*, chromosome-level genome, comparative genomics, biological evolution

## Abstract

**Simple Summary:**

In this study, a high-quality chromosomal-level genome of male *Decapterus maruadsi* was assembled based on Illumina, PacBio, and Hi-C technology. Notably, 23 chromosome-level genome sequences with lengths ranging between 21.74 and 44.53 Mb were assembled. A total of 22,716 protein-coding genes with an average transcript and a CDs length of 12,823.03 bp and 1676.66 bp, respectively, were successfully annotated. Based on positive selection analysis, some genes associated with the growth and development of bone, muscle, cardioid, and ovary were screened. These genes were likely involved in the evolution of typical biological features in *D. maruadsi*, such as fast growth rate, small body size, and strong fecundity. The newly established reference genome provides a fundamental genome resource for further genetic conservation, genomic-assisted breeding, and exploration of the molecular mechanism underlying adaptive evolution.

**Abstract:**

*Decapterus maruadsi* is a typical representative of small pelagic fish characterized by fast growth rate, small body size, and high fecundity. It is a high-quality marine commercial fish with high nutritional value. However, the underlying genetics and genomics research focused on *D. maruadsi* is not comprehensive. Herein, a high-quality chromosome-level genome of a male *D. maruadsi* was assembled. The assembled genome length was 716.13 Mb with contig N50 of 19.70 Mb. Notably, we successfully anchored 95.73% contig sequences into 23 chromosomes with a total length of 685.54 Mb and a scaffold N50 of 30.77 Mb. A total of 22,716 protein-coding genes, 274.90 Mb repeat sequences, and 10,060 ncRNAs were predicted, among which 22,037 (97%) genes were successfully functionally annotated. The comparative genome analysis identified 459 unique, 73 expanded, and 52 contracted gene families. Moreover, 2804 genes were identified as candidates for positive selection, of which some that were related to the growth and development of bone, muscle, cardioid, and ovaries, such as some members of the TGF-β superfamily, were likely involved in the evolution of typical biological features in *D. maruadsi*. The study provides an accurate and complete chromosome-level reference genome for further genetic conservation, genomic-assisted breeding, and adaptive evolution research for *D. maruadsi*.

## 1. Introduction

The Japanese scad *Decapterus maruadsi* (Temminck and Schlegel, 1843) is a small pelagic fish common to warm-temperate coastal areas and belongs to the order Perciformes, family Carangidae [1]. The Japanese scad is widely distributed across the marginal sea of the Indo-West Pacific and is an important economic fish in Asian coastal countries such as China, Malaysia, and Thailand [2,3,4]. For example, the annual catch of *D. maruadsi* in China reached approximately 5.5 × 10^5^ tons from 1996 through 2023 (Chinese Fishery Statistical Yearbook, 1997~2023) [5]. Notably, *D. maruadsi* has been classified as overfished in some coastal sea areas, such as the coastal waters of the Beibu Gulf (annual exploitation rate = 0.63~0.78) and southern Zhejiang (annual exploitation rate = 0.61) in China [6,7]. At the same time, the phenotypic characteristics, population structure, and heritable traits of *D. maruadsi* have revealed a population decline, including miniaturization, precocious puberty, structure simplification, and low levels of genetic diversity (nucleotide diversity *π* < 0.005) [2,3,4,8,9]. The Japanese scad should thus be closely monitored to ensure the sustainable utilization and scientific management of fishery resources.

The Japanese scad is an R-selected fish characterized by strong environmental adaptability, wide distribution range, short life cycle, rapid growth rate, simple population structure (consists of 0–5 age, and 1–2 age predominates among them), rapid generational renewal, early sexual maturity, strong individually absolute fecundity, long egg-laying period, and wide spawning grounds [10]. Wild *D. maruadsi* is relatively small in size relative to *Caranx melampygus*, *Seriola lalandi*, *Seriola dumerili*, *Thunnus albacares*, *Thunnus maccoyii*, and other large marine fish. Bottom trawl survey data revealed that the body length of *D. maruadsi* was 32~300 mm from 1992 to 2012 in the Beibu Gulf [11], while the body length and weight were 78~248 mm and 7.9~290.0 g from 2014 to 2017 in the northern South China Sea, respectively [8]. Wild *D. maruadsi* has some considerable migratory ability, but its ability is significantly lower than that of highly migratory marine fish, such as *T. albacares* and *T. maccoyii*. In addition, different *D. maruadsi* populations have varying migratory abilities. For example, *D. maruadsi* populations in the northern South China Sea only migrate from the deep-water areas of the outer sea to the nearshore for a short-distance spawning migration. In contrast, the populations in the East China Sea migrate from the central and southern Taiwan Strait and northern Taiwan to the Zhejiang Province coastal waters for long-distance spawning and feeding migration [2,10,12]. Generally, small pelagic fishes, such as wild *D. maruadsi*, anchovy, and sardine, are characterized by fast growth, small size, and high fecundity. However, the genetic basis in these characteristic formations has rarely been reported so far.

The Japanese scad is a high-quality marine commercial fish with high nutritional value. The muscle is rich in ideal protein (essential amino acids accounting for 39.98% of total amino acids), unsaturated fatty acids (UFA, accounting for 60.79% of total fatty acids), and various trace elements essential to the human body [13]. Currently, the *D. maruadsi* aquaculture industry has been scientifically developing, and *D. maruadsi* has been able to be artificially reproduced successfully in Dongshan County, Fujian Province, China. Cultured *D. maruadsi* are not high-yielding but are large, grow fast, have a tender taste, and have a high nutrition value (UFAs accounting for 71.44% of total fatty acids, essential amino acids accounting for 43.45% of total amino acids, both being significantly higher than those of wild fish) [14]. Moreover, cultured *D. maruadsi* can be used to prepare sashimi because of their delicious taste and high nutritional value, and the price is much higher than that of wild *D. maruadsi*. Cultured *D. maruadsi* thus achieves a high-value application and is worth further development and promotion. To ensure the high-quality development of *D. maruadsi* aquaculture, it is crucial to investigate the germplasm resources and to breed excellent varieties. This requires a large amount of basic research and high-quality reference genomes, which are essential basic data and tools for the sustainable usage and superior breeding varieties of *D. maruadsi*.

Notably, the existing basic *D. maruadsi* research does not match its high economic value and rapid aquaculture development. Previous studies on *D. maruadsi* are not comprehensive and mainly focus on the biological characteristics, investigation and evaluation, population genetics, and feeding habits of wild resources. Relative to other cultured fish species, the underlying genetic basis of *D. maruadsi* is reported in small amounts. Zhao [15] obtained 1539.7 Mb of the *D. maruadsi* genome using second-generation sequencing technology. However, the assembled genome was too long because of the numerous short reads, assembly difficulties, and error regions and gaps generated in the assembly process. Chen et al. [16] reported a chromosome-level reference genome of female *D. maruadsi* based on Nanopore sequencing and Hi-C technology, and the final genome size was 713.58 Mb and anchored on 23 chromosomes. Li et al. [17] obtained a 16,541bp mitochondrial genome of *D. maruadsi* using overlapped PCR. The genetic structure and diversity of *D. maruadsi* from Chinese coastal waters and northern Vietnam were evaluated based on mtDNA *COI*, Cyt *b*, and control region sequences [3,9,18,19,20]. Hou et al. [21] revealed spatial and temporal information on the distributions of *D. maruadsi* eggs with DNA barcodes in the northern South China Sea. Nonetheless, further genetic research on *D. maruadsi* is beneficial for better promoting the comprehensive analysis of the genetic mechanism behind the biology (body size, growth rate, and fertility), selection of good varieties and the comprehensive assessment of germplasm resources.

In this study, second-generation Illumina, three-generation PacBio, and Hi-C techniques were used to construct a chromosome-level genome of male *D. maruadsi*. The repeated sequences, protein-coding genes, and noncoding RNA (ncRNA) of the genome were annotated based on the RNA-seq data by Illumina and PacBio sequencing. Comparative genomic analysis was used to determine the phylogenetic relationship and divergence time of 17 fish species, screen the candidate genes associated with typical biological characteristics of *D. maruadsi*, and identify the collinear regions between the genome assembled in this study and the other two genomes. The findings of this study provide an accurate chromosome-level genome for the analysis of the genetic base of economic traits and genome selection breeding in *D. maruadsi*. The findings also contribute to the conservation of germplasm resources and the analysis of the molecular mechanisms of adaptive evolution for *D. maruadsi* and other important small pelagic fish based on their whole genomes.

## 2. Materials and Methods

### 2.1. Sample Collection

Three *D. maruadsi* samples (Figure 1) were collected from the Zhanjiang Bay of the South China Sea (Zhanjiang City, Guangdong Province, China) for genome sequencing. The body length and weight were measured after anesthetizing the fish with MS-222, followed by harvesting of the muscle, liver, and heart tissues in liquid nitrogen and the subsequent storage of the tissues at ultra-low temperature (−80 °C). Part of the muscle tissue was used for genomic DNA sequencing, while the muscle, liver, and heart tissues were used for transcriptome sequencing. The experimental animal protocols used in this study were approved by the Animal Ethics Committee of Guangdong Ocean University, China.

### 2.2. DNA Library Construction and Sequencing (Illumina and PacBio)

Genomic DNA was isolated from the muscle tissues using the phenol-chloroform extraction method [22]. The degradation and contamination of the DNA samples were checked on a 1% agarose gel electrophoresis. DNA purity and concentration were measured using a NanoPhotometer spectrophotometer (IMPLEN, Westlake Village, CA, USA) and a Qubit 2.0 Fluorometer (Life Technologies, Carlsbad, CA, USA), respectively. The qualified DNA was used to construct second-generation and third-generation sequencing libraries.

A short-fragment DNA library (350 bp) was constructed using the Truseq Nano DNA HT Sample Preparation Kit (Illumina, San Diego, CA, USA). Briefly, the qualified DNA was randomly fragmented using a Covaris ultrasonic crusher, followed by the preparation of a short-fragment DNA library through terminal repair, A-tail addition, sequencing adaptor addition, purification, and PCR amplification. The qualified libraries were sequenced on an Illumina NovaSeq6000 platform (Illumina, San Diego, CA, USA). The raw sequence reads were fine-filtered to obtain high-quality clean reads by removing the adaptor sequences, reads with more than 10% unknown bases (N), and single-end reads with low-quality bases greater than 20%. The sequence error rate, GC Content, Q20, and Q30 of the clean reads were then calculated to assess the sequence quality. Ten thousand randomly selected clean reads (read1 and read2 each 5000) were mapped to the NCBI-NT database by BLAST to check and exclude the exogenous contaminants.

A SMRTbell library (15 k, PacBio library) was constructed using a SMRTbell Express Template Prep Kit 2.0 (Pacific Biosciences, Menlo Park, CA, USA). The qualified DNA was first randomly fragmented using a Covaris ultrasonic crusher, followed by the enrichment and purification of the large DNA fragments (5~20 kb) using Ampure PB beads. The Template Prep Kit was then used to do damage and end repair, followed by ligation of the fragments with circular sequencing adapters. The exonucleases were finally used to remove the failed ligation DNA fragments, and the library quality was inspected using Femto Pulse. The qualified SMRTbell library was sequenced on a PacBio Sequel II platform (Pacific Biosciences, Menlo Park, CA, USA).

### 2.3. De Novo Genome Assembly and Quality Assessment

K-mer frequency analysis was used to estimate the genome size, heterozygous ratio, and repeat ratio of the Illumina sequencing data using Jellyfish v2.2.7 [23]. SMRTlink v10.2 software [24] was used to filter PacBio sequencing raw data to remove low-quality reads and obtain high-quality HiFi reads. The PacBio HiFi reads were subsequently assembled to obtain a draft genome using the Hifiasm v0.15.4 software [25]. The completeness of the assembled genome was assessed by Bench-marking Universal Single-Copy Orthologs (BUSCO) and the Core Eukaryotic Genes Mapping Approach (CEGMA) [26,27]. Qualified Illumina DNA sequencing data were mapped onto the assembled genome to calculate the mapping ratio of the clean reads and the degree and depth of genome coverage using the BWA v0.7.8 software [28], which also assessed the completeness of genome assembly and the uniformity of sequencing. Single nucleotide polymorphism (SNP) calling was performed using Samtools v 0.1.19 [29] based on the BWA mapping results. The homozygous and heterozygous SNPs were identified to assess the accuracy of the genome assembly. The GC content of the assembled genome was calculated using 10 kp non-overlapping sliding windows. A GC content distribution analysis was done to detect the AT and GC separation of the sequencing data.

### 2.4. Chromosome-Level Genome Assembly

The high-throughput chromosome conformation capture (Hi-C) technique [30,31] was used to construct the chromosome-level genome assembly for *D. maruadsi*. A Hi-C library was first constructed. Cells were fixed with paraformaldehyde and lysed, followed by digesting the crosslinked DNA with restriction enzymes to obtain sticky ends. The ends of the DNA fragments were repaired, labelled with biotin, and ligated again. The cross-linked DNA was digested with proteinase, purified, and randomly sheared to 300~500 bp. The biotin-containing DNA fragments were captured by avidin beads. The purified DNA fragments were subjected to end-repair, A-tailing, sequencing adaptor ligation, PCR amplification, and purification for Hi-C library construction. The completeness of the DNA fragments and inserted fragment sizes were detected using Agilent 2100 (Agilent Technologies, Palo Alto, CA, USA). The effective library concentration was quantified using Qubit 2.0 and quantitative PCR (qPCR). Different qualified libraries were pooled based on the requirements of effective concentration and target data volume and were then sequenced on an Illumina PE150 platform. The quality of the Hi-C sequencing data was checked using a statistical comparison method (described in 2.2) and HiCUP quality control analysis to obtain effective whole-genome chromosome crosslinking information [32]. The clean Hi-C reads were finally mapped to draft genome contigs using ALLHiC v0.9.8 [33] to obtain a chromosome-level genome of *D. maruadsi*.

### 2.5. Genome Annotation

#### 2.5.1. RNA Library Construction and Sequencing (Illumina and PacBio)

A mixed RNA sample from muscle, liver, and heart tissues was prepared for PacBio full-length sequencing based on single-molecule, real-time (SMRT) sequencing technology to obtain an accurate, full-length transcript and to precisely annotate the genome. Each sample was also subjected to high-throughput sequencing on an Illumina NovaSeq6000 platform. The total RNA was extracted from muscle, liver, and heart tissues using the Trizol Reagent Kit (Invitrogen, Carlsbad, CA, USA). The Agilent Bioanalyzer 2100 and Nanodrop 2000 Spectrophotometer (Thermo Fisher Scientific, Waltham, MA, USA) were used to detect the completeness, concentration, and purity of the total RNA. Good-quality RNA samples were subsequently used for library preparation and sequencing.

The total RNA isolated from the three tissues was mixed in equal amounts, and the mRNA was enriched using oligo (dT) magnetic beads. The full-length cDNA was synthesized using a SMARTer PCR cDNA Synthesis Kit (Takara Bio, Beijing, China). The cDNA was then subjected to PCR amplification, terminal repair, and the attachment of dumbbell-shaped SMRT adapters to construct a full-length transcriptome library. The library quality was checked, and the good-quality libraries were sequenced on a PacBio Sequel sequencer (Pacific Biosciences). The subread sequences were obtained by processing the raw data on SMRTlink (PacBio) and correcting it to obtain circular consensus sequences (CCS). The sequences were divided into full-length sequences and non-full-length sequences based on whether the CCS contained 5′-end primer, 3′-end primer, or polyA tail. The full-length sequences were clustered to obtain the clustered consensus sequences. The accurate, full-length transcripts were finally obtained using isoseq3 in SMRTlink software (https://www.omicsclass.com/article/344, accessed on 26 September 2023).

The mRNA was enriched using oligo (dT) magnetic beads and fragmented in a fragmentation buffer. The first-strand cDNA was synthesized using random hexamer primers, while the second-strand cDNA was synthesized using DNA polymerase I and RNase H. The cDNA was purified and subjected to end repair, poly(A) addition, and Illumina sequencing adaptor ligation. Those with suitable sizes were selected for PCR amplification. The PCR products were subsequently purified using AMPure XP Beads to obtain a short-fragment sequencing library. Good-quality libraries were then sequenced on an Illumina NovaSeq 6000 platform (Illumina, San Diego, CA, USA). Clean reads were obtained by the fine filtering of raw reads using Fastp v0.21.0 [34].

#### 2.5.2. Genome Annotation

The chromosome-level genome and full-length transcript sequences were used to annotate repetitive sequences, genes (structure and function), and ncRNA. Repetitive sequences were annotated using homologous sequence alignment and de novo prediction. Sequences similar to the known repetitive sequences were predicted using the RepBase database, Repeatmasker v3.3.0 [35], and Repeatproteinmask v4.05 [36]. LTR_Finder v1.07 [37], RepeatScout v1.05 [38], and RepeatModeler v1.0.5 were employed to build a de novo repetitive sequences library. Repeatmasker v4.1.2 and TRF v4.07b [39] were used to predict de novo repetitive sequences and tandem repeat sequences, respectively. The repetitive sequences predicted by the two methods were merged and annotated using Repeatmasker v4.1.2.

Protein-coding genes were predicted by combining de novo prediction, homologous prediction, and RNA-seq-assisted annotation methods. Augustus v3.2.3 [40], Genescan v1.0 [41], Geneid [42], GlimmerHmm v3.0.2 [43], and SNAP [44] were used for de novo prediction. *D. maruadsi* genome sequences were aligned to the homologous protein-coding sequence of 11 species, including *Caranx melampygus*, *Seriola dumerili*, *Seriola lalandi*, *Epinephelus lanceolatus*, *Epinephelus akaara*, *Plectropomus leopardus*, *Danio rerio*, *Takifugu rubripes*, *Gasterosteus aculeatus*, *Oryzias latipes*, and *Homo sapiens*, using TBLASTN v2.2.26 [45] and Genewise v2.2.0 [46]. The RNA-Seq data were aligned to the reference genome of *D. maruadsi* (obtained in this study) using TOPHAT v2.0.8 [47] and Cufflinks v2.1.1 [48] to assist in predicting the gene structure. All predicted genes were merged using the EvidenceModeler (EVM) v1.1.1 [49] to form a non-redundant and comprehensive gene set. Based on the full-length transcript sequences obtained in 2.5, the annotation results of EVM for the final gene sets were corrected to add UTR and variable splicing information using PASA. These predicted genes subsequently underwent functional and structural annotations using Blastp v2.2.29+ [50] and HMMER v3.1b2 [51] on the Swiss-Prot Protein Knowledgebase (SwissProt), the Non-Redundant Protein (Nr), Kyoto Encyclopedia of Genes and Genomes (KEGG), and the Integrated Resource of Protein Domains and Functional Sites (InterPro).

Four types of ncRNA, including tRNA, rRNA, miRNA, and snRNA, were identified. tRNAs were predicted using tRNAscan-SE v2.0 [52]; rRNAs were predicted by BLAST, based on the rRNA sequences of closely related species, while miRNA and snRNA were predicted by Infernal v1.1rc4 [53] from Rfam v1.1.4 [54].

### 2.6. Comparative Genome Analysis

Cluster analysis of gene families in 17 fish genomes was performed using OrthoMCL v1.4 (http://orthomcl.org/orthomcl/, accessed on 6 November 2023) [55] to identify single-copy gene families, multicopy gene families, and species-unique genes and gene families. The 17 fish included *Acanthopagrus schlegelii*, *Ctenopharyngodon idella*, *Cetorhinus maximus*, *C. melampygus*, *D. maruadsi*, *D. rerio*, *E. akaara*, *E. lanceolatus*, *Hypophthalmichthys molitrix*, *Larimichthys crocea*, *O. latipes*, *Oreochromis niloticus*, *Perca flavescens*, *S. dumerili*, *S. lalandi*, *Thunnus albacares*, and *Thunnus maccoyii*. The expansion and contraction of gene families were analyzed using CAFÉ v4.0 [56]. The expanded and contracted genes were then subjected to GO and KEGG enrichment analyses to further analyze the genetic changes behind phenotypic divergence between the fish species.

The single-copy gene families (2468) shared by the 17 fish species were aligned using MUSCLE [57], and the resulting alignment (Appendix A) was used to construct a maximum likelihood phylogenetic tree using RAxML v8.2.12 software [58] with 1000 bootstrap replicates and *Cetorhinus maximus* as the outgroup (GCF_000165045.1). The divergence time among the different species was estimated using MCMCTree [59] from the PAML package and corrected using the following time correction points (TimeTree database, http://www.timetree.org/, accessed on 13 November 2023): *L. crocea* vs. *A. schlegelii* (102~127 Mya), *C. melampygus* vs. *S. dumerili* (49~66 Mya), *O. latipes* vs. *L. crocea* (104~145 Mya), *D. rerio* vs. *C. idella* (48~75 Mya), *D. rerio* vs. *L. crocea* (206~252 Mya), and *C. maximus* vs. *L. crocea* (453~497 Mya).

Candidate genes associated with the body size, growth velocity, and migratory habits of *D. maruadsi* were screened using three groups of positive selection analyses based on single-copy gene families: D. *maruadsi* vs. *C. melampygus*, *S. dumerili* and *S. lalandi* (group 1); *D. maruadsi* vs. *T. albacares* and *T. maccoyii* (group 2); and *D. maruadsi* vs. *L. crocea*, *E. akaara* and *A. schlegelii* (group 3). Multiple protein sequence alignments of single-copy gene families were conducted using MUSCLE v3.7 [57], and the alignment results were used as templates to generate the CDs alignment results. The codeml program in PAML was employed to test whether the genes were under positive selection using the branch-site specific model. GO and KEGG enrichment analyses were performed to obtain candidate genes associated with growth, body size, and migratory habits from the positive selection genes.

The NGenomeSyn v1.41 software [60] was used to construct the syntenic blocks based on the high-quality chromosome-level genomes of male *D. maruadsi* (this study), female *D. maruadsi* (GCA_030347415.2, 723.8 Mb of genome size, 13.6 Mb of contig N50, 32.2 Mb of scaffold 50), and male *T. maccoyii* (GCF_910596095.1, 782.4 Mb of genome size, 26.8 Mb of contig N50, 33.8 Mb of scaffold 50). This analysis was done to assess the accuracy and completeness of the assembled genome and identify the genome homologous regions between the assembled genome and other genomes.

## 3. Results

### 3.1. Analysis of Genomic Characterization

In this study, 118,505,182 raw paired reads and 35,551,554,600 bp (35.55 G) raw base data were generated by Illumina DNA sequencing (Table 1). The removal of low-quality data yielded 91,050,982 clean paired reads and 27,315,294,600 bp clean data, with 96.48% of Q20, 91.15% of Q30, 0.04% of sequencing error rate, and 42.58% GC content. These values illustrated that the construction and sequencing quality of the genomic DNA short-fragment library was good and would guarantee accuracy in the subsequent analyses. Ten thousand clean reads of *D. maruadsi* were randomly selected and mapped to the nucleotide sequences (NT) database. Notably, the top six species with the highest sequence coverages were *Dicentrarchus labrax* (0.53%), *Haplochromis burtoni* (0.45%), *O. niloticus* (0.31%), *Xiphophorus maculatus* (0.19%), *T. rubripes* (0.19%), and *G. aculeatus* (0.16%), indicating that the alignments were orthologous and the *D. maruadsi* sample was not contaminated with external nucleotides. The genome of *D. maruadsi* was estimated to be approximately 739.40 Mbp (Appendix A), with 1.18% heterozygosity and 35.16% repeat sequences, based on the filtered sequence data (clean reads). The genome was subsequently revised to 722.79 Mbp using K-mer frequency analysis (accession number JBANGS000000000).

### 3.2. Genome Assembly and Evaluation

A total of 23,164,121,378 bp (23.16 G) sequencing data and 1,765,660 high-quality HiFi reads were generated by PacBio-SMRT sequencing. The average length and N50 length of the HiFi reads were 13,119 bp and 13,207 bp, respectively, and the sequencing depth was 32.04× (based on the estimated genome size of 722.79 M by survey). The 1,765,660 high-quality HiFi reads were assembled into 349 contigs that were further error-corrected based on Illumina sequencing data (27.32 G). Finally, 716,127,322 bp (716.13 Mb) of the draft genome with 20,796,328 bp of N50 were obtained. Notably, this genome size was very close to the estimated genome size based on K-mer frequency analysis (722.79 Mb).

BUSCO analysis (Appendix A) yielded 3538 (97.2%) complete BUSCOs, of which 3494 (96.0%) were complete single-copy BUSCOs and 44 (1.2%) were complete duplicated BUSCOs. This finding suggested that the *D. maruadsi* genome assembled based on PacBio sequencing data had high coverage of all gene regions. CEGMA evaluation revealed that the assembled genome of *D. maruadsi* completely covered 233 of 248 (93.95%) conserved-core eukaryotic genes. The evaluation results of BUSCO and CEGMA strongly suggested that the assembled genome sequence of *D. maruadsi* was relatively complete.

All clean reads obtained by Illumina sequencing were mapped onto the assembled genome. The mapping rate, coverage rate, and average per-base sequencing depth were approximately 97.76%, 99.94%, and 36.05%, respectively, indicating excellent consistency between the Illumina reads and the assembled genome. The alignments yielded 5,341,607 SNPs (0.7547%), including 5,340,262 heterozygous SNPs (0.7545%) and 1345 homozygous SNPs (0.0002%), indicating that the assembled genome had a high single-base accuracy. The GC content of the assembled genome was concentrated around 42.49%, and there was no obvious GC separation, indicating that there was no exogenous pollution in the genome.

### 3.3. Chromosome-Level Genome Assembly by Hi-C

The draft genome of *D. maruadsi* was further scaffolded using Hi-C technology. Illumina sequencing (Hi-C library presequencing data) generated 12,383,804 raw paired reads and 3,715,141,200 bp (3.7 G) raw data. A total of 10,430,159 clean paired reads and 3,708,356,100 bp clean data were generated after quality control. The sequence quality values Q20 and Q30 were 90.75% and 96.34%, respectively, while the sequencing error rate and the GC content were 0.04% and 44.14%, respectively. These indicators suggested that the Hi-C library construction and sequencing were of high quality. HiCUP analysis (Appendix A) revealed that 6,544,378 of the 10,430,159 clean paired reads were successfully matched (62.74% total paired ratio), 5,564,284 (85.02%) were valid read pairs (Di-tags), and 980,094 (14.98%) were invalid Di-tags (same circularized, same fragment dangling ends, same fragment internal). A total of 5,176,451 unique valid Di-tags were generated after the duplicate Di-tags in the valid Di-tags were filtered (Appendix A). The unique valid Di-tags included 2,393,982 cis Di-tags (489,982 cis-close Di-tags and 1,904,000 cis-far Di-tags) and 2,782,469 trans-Di-tags. In summary, the effective utilization rate of Hi-C data was 49.63% (unique, valid Di-tags/clean paired reads), indicating that the Hi-C library construction, sequencing, and analytical results were valid. Hi-C library can thus be used for massive sequencing to derive chromosome-level genome assembly.

The total sequencing data of Hi-C showed that a total of 251,272,317 raw paired reads and 75,381,695,100 bp (75.38 G) raw data were generated by the Illumina sequencing platform. Initial quality control yielded 208,470,664 clean paired reads and 62,541,199,200 bp clean data. The Q20, Q30, sequencing error rate, and GC content were 96.30%, 90.91%, 0.04%, and 44.23%, respectively, indicating high-quality Hi-C library construction and sequencing. Ten thousand clean reads were randomly selected for comparison with the NT database. The top six species with the highest sequence coverages were *H. burtoni* (0.41%), *O. niloticus* (0.37%), *D. labrax* (0.37%), *Trachurus japonicus* (0.17%), *D. maruadsi* (0.14%, second-generation sequencing assembly), and *X. maculatus* (0.12%), showing that *D. maruadsi* was highly homologous with these fish and its draft genome was not contaminated with external nucleotides.

The draft genome assembled in Section 3.2 was loaded onto the chromosomes based on the valid Hi-C data after quality control (Table 2). A total of 357 contigs (≥100), with 716,127,322 bp of the total length and 19,703,568 bp of N50, were generated. These contigs were further assembled into 74 scaffolds with 716,155,622 bp of the total length and 30,768,099 bp of N50. Among the 74 scaffolds, 23 scaffolds with a total length of 685,544,437 bp were assembled into 23 chromosome-level sequences, while the remaining 51 scaffolds with a total length of 30,611,185 bp were not assembled into chromosome-level sequences (Figure 2). The final assembly spanned 23 chromosomes with sizes ranging between 21.74 and 44.53 Mb, representing 95.73% of the genome.

### 3.4. Genome Annotation

#### 3.4.1. PacBio and Illumina RNA-Seq Data

A total of 62,936,860 raw reads and 18.88 G raw data were obtained through transcriptome sequencing of the muscle, liver, and heart tissues. The removal of low-quality reads yielded 60,532,297 clean reads and 18.17 G clean data. The clean reads drawn from the muscle, liver, and heart tissues were 20,827,273, 18,887,824, and 20,817,200, respectively (Appendix A). The clean data drawn from the muscle, liver, and heart tissues were 6.25 G, 5.67 G, and 6.25 G, respectively. The Q20, Q30, GC content, and sequencing error rate of the clean reads of the three tissues were 97.69%~97.83%, 93.54%~93.85%, 49.80%~52.83%, and 0.03%, respectively, which suggested that the Illumina sequencing quality was good.

Full-length transcriptomes of the mixed samples of the three tissues were also obtained after PacBio SMRT sequencing. A total of 835,845 polymerase reads and 83.33 G polymerase read bases were obtained, with a mean length of 99,697 bp and an N50 of 170,923 bp. These polymerase reads were split into 24,423,333 subreads (81.52 G) with an average length of 3338 bp and N50 of 3778 bp. The high-quality transcriptome data based on Illumina and PacBio sequencing were subsequently used to assist in genome annotation.

#### 3.4.2. Prediction of Repetitive Sequences

In total, 274,895,699 bp repetitive sequences, accounting for 38.39% of the whole genome, were identified (Appendix A). Among them, 72,744,333 bp (10.16%) were tandem repeats, while 202,151,366 bp (28.23%) were interspersed repeats (Figure 3). The interspersed repeat sequences included 82,090,490 bp (11.46%) DNA transposons, 101,257,190 bp (14.14%) retrotransposon, 55 bp (0.000008%) other transposons, and 5,046,848 bp (0.70%) unknown sequences. The retrotransposon included 37,075,462 bp (5.18%) long interspersed nuclear elements (LINE), 1,928,796 bp (0.27%) short interspersed nuclear elements (SINE), and 62,252,932 bp (8.69%) long terminal repeat retrotransposons (LTR).

#### 3.4.3. Structural and Functional Annotation of Protein-Coding Genes

A total of 27,885 protein-coding genes were annotated in the genome of *D. maruadsi* by combining homology-based, de novo, and RNA-seq-assisted prediction methods (Table 3). A total of 22,716 protein-coding genes with UTR regions were obtained after the variable shear, low-quality transcripts (overlapping with TE ≥ 20%, premature termination, only de novo evidence supported, less than one of rpkm expression in all tissues), and redundant single-exons were removed (Figure 4A). The average lengths of the protein-coding genes and coding region were 12,823.03 bp and 1676.66 bp, respectively. Each gene contained an average of 9.65 exons. The average lengths of exons and introns were 173.83 bp and 1289.29 bp, respectively. Subsequently, 22,716 protein-coding genes were functionally annotated using SwissProt, Nr, KEGG, and InterPro databases (Figure 4B). Finally, a total of 22,037 (97%) genes were functionally annotated in at least one database, while the remaining 679 (3%) genes were unannotated.

#### 3.4.4. ncRNA Annotation

In total, 10,419 ncRNAs, including 1829 miRNAs (total length 234,126 bp, 0.0327%), 2842 tRNAs (total length 214,675 bp, 0.0300%), 5310 rRNAs (total length 598,939 bp, 0.0836%), and 438 snRNAs (total length 57,170 bp, 0.00798%) were annotated in the *D. maruadsi* genome (Appendix A). The rRNAs contained all of the four major rRNA components: 1809 of 18S rRNA, 82 of 28S rRNA, 2 of 5.8S rRNA, and 3417 of 5S rRNA.

### 3.5. Comparative Genome Analysis

#### 3.5.1. Gene Family Clustering, Expansions, and Contractions

Cluster analysis of 16,858 (*C. maximus*) and ~32,712 (*C. idella*) genes in *D. maruadsi* and 16 other fish revealed 23,981 gene families, among which 2468 were common single-copy gene families (Figure 5A and Table 4). A Venn diagram of the gene families (Figure 5B) showed that 11,871 orthologous gene families were shared between *D. maruadsi* and three other fish species of the same family Carangidae (*C. melampygus*, *S. dumerili*, and *S. lalandi*), while 459 gene families were unique to *D. maruadsi*. GO enrichment analysis (Appendix A) of the 459 unique gene families categorized them into 32 GO terms, including RNA-directed DNA polymerase activity (GO:0003964), RNA-dependent DNA biosynthetic process (GO:0006278), and carbohydrate-binding (GO:0030246), among other GO terms. KEGG enrichment analysis showed that the unique gene families were mainly involved in the regulation of lipolysis in adipocytes (map04923), the PPAR signaling pathway (map03320), the synaptic vesicle cycle (map04721), glycosaminoglycan biosynthesis—heparan sulfate/heparin (map00534), the Apelin signaling pathway (map04371), peroxisome (map04146), the biosynthesis of unsaturated fatty acids (map01040), and the GABAergic synapse (map04727), among other functions and pathways.

The expansion and contraction analysis of 23,981 gene families in 17 species (Figure 6) showed that 73 gene families were expanded while 52 gene families were contracted in *D. maruadsi* compared with the common ancestors of *D. maruadsi* and *C. melampygus* during the evolutionary process. Enrichment analysis (Appendix A) showed that the 73 expanded gene families were categorized into 34 GO terms, including exo-alpha-sialidase activity (GO:0004308), nucleosome (GO:0000786) and nucleosome assembly (GO:0006334), and 27 KEGG pathways, including sphingolipid metabolism (map00600), PPAR signaling pathway (map03320), primary bile acid biosynthesis (map00120), longevity regulating pathway—multiple species (map04213), and regulation of lipolysis in adipocytes (map04923), among other functions and pathways. In the same line, the 52 contracted (Appendix A) gene families were mainly involved in 27 GO terms, including neurotransmitter: sodium symporter activity (GO:0005328), neurotransmitter transport (GO:0006836), and homophilic cell adhesion via plasma membrane adhesion molecules (GO:0007156), as well was 22 KEGG pathways, including synaptic vesicle cycle (map04721), GABAergic synapse (map04727), graft-versus-host disease (map05332), allograft rejection (map05330), viral myocarditis (map05416), among other functions and pathways. Notably, the unique gene families and the expansion and contraction gene families of *D. maruadsi* are involved in the PPAR signaling pathway, the regulation of lipolysis in adipocytes, the synaptic vesicle cycle, and GABAergic synapse, amongst other functions.

#### 3.5.2. Phylogenetic Tree and Divergence Times

A phylogenetic tree constructed based on 2468 single-copy gene families shared by the 17 fish species showed that *D. maruadsi* and *C. melampygus* clustered into one clade with 100% bootstrap support and then clustered with two other Carangidae fish (*S. lalandi* and *S. dumerili*) with a bootstrap support of 100%. Four Carangidae fish were grouped with other seven Perciform fish and finally clustered with one Perciform fish, one Beloniform fish, and three Cypriniform fish. This result is not entirely consistent with the taxonomic classification based on their morphological characteristics. Mismatches between morphological and molecular identifications are common for the classification of many organisms, and more powerful phylogenetic evidence will be needed for the accurate classification. The evolutionary tree of the 17 fish species (Figure 7) showed that the divergence between *D. maruadsi* and *C. melampygus* occurred approximately 36.4 (26.0~48.1) million years ago. The other 14 teleost were divergent from 6.7 million years to 242.7 million years. The phylogeny and divergence times can help us better identify and understand the evolutionary history of teleost in the future.

#### 3.5.3. Positive Selection Analysis

A total of 1233 candidate genes under positive selection (FDR ≤ 0.05) were identified in the first group of positive selection analysis based on the likelihood ratio test (*D. maruadsi* vs. *C. melampygus*, *S. dumerili*, and *S. lalandi*). The candidate genes were significantly enriched in 58 GO terms (Figure 8 and Table 5), including the thrombin-activated receptor signaling pathway (GO:0070493), syntaxin binding (GO:0019905), syntaxin-1 binding (GO:0017075), signaling receptor binding (GO:0005102), rough endoplasmic reticulum membrane (GO:0030867), and receptor ligand activity (GO:0048018), among other GO terms. KEGG enrichment analysis revealed that the 1233 candidate genes were significantly enriched in 17 pathways, including primary immunodeficiency (map05340), the JAK-STAT signaling pathway (map04630), hematopoietic cell lineage (map04640), cytokine-cytokine receptor interaction (map04060), and complement and coagulation cascades (map04610), among other functions and pathways.

A total of 810 candidate genes under positive selection (FDR ≤ 0.05) were identified in the second group of positive selection analysis (*D. maruadsi* vs. *T. albacares* and *T. maccoyii*). These candidate genes were significantly enriched in 56 GO terms and 16 KEGG pathways (Figure 8 and Table 5). GO terms mainly included polysaccharide binding (GO:0030247), nucleotide-excision repair (GO:0006289), nucleic acid binding (GO:0003676), nuclease activity (GO:0004518), and immune system process (GO:0002376). KEGG pathways mainly included the JAK-STAT signaling pathway (map04630), intestinal immune network for IgA production (map04672), hematopoietic cell lineage (map04640), cytokine-cytokine receptor interaction (map04060), and complement and coagulation cascades (map04610).

In total, 761 candidate genes under positive selection (FDR ≤ 0.05) were identified in the third group of positive selection analysis (*D. maruadsi* vs. *L. crocea*, *E. akaara*, and *A. schlegelii*). These candidate genes were mainly enriched in 48 GO terms (Figure 8, Table 5), including telomerase holoenzyme complex (GO:0005697), the structural constituent of ribosome (GO:0003735), strictosidine synthase activity (GO:0016844), ribosome (GO:0005840), ribokinase activity (GO:0004747), and the regulation of catabolic process (GO:0009894). These candidate genes were significantly enriched in nine KEGG pathways, including the PI3K-Akt signaling pathway (map04151), the p53 signaling pathway (map04115), the JAK-STAT signaling pathway (map04630), hematopoietic cell lineage (map04640), cytokine-cytokine receptor interaction (map04060), and complement and coagulation cascades (map04610).

#### 3.5.4. Collinearity Analysis

The 23 chromosomes of the male *D. maruadsi* displayed significant collinearity with 23 chromosomes of the female *D. maruadsi* and 24 chromosomes of the male *T. maccoyii*, a closely related species in the Carangidae family (Figure 9 and Table 6). Notably, the genomic collinearity between the male *D. maruadsi* and the male *T. maccoyii* outperformed that between the male *D. maruadsi* and the female *D. maruadsi*. Moreover, chromosome 14 of the male *D. maruadsi* corresponded strongly to chromosome 7 and chromosome 24 of the male *T. maccoyii*. These results collectively demonstrated the high accuracy, completeness, and continuity of the genomes assembled in this study.

## 4. Discussion

### 4.1. Genome Features

The chromosome-level genome of the male *D. maruadsi* were assembled based on Illumina, PacBio, and Hi-C technologies. Notably, the assembled genome size of the male *D. maruadsi* is 716.13 Mb at the contig level. Finally, 23 chromosome-level genome sequences with lengths ranging between 21.74 and 44.53 Mb were assembled, and the total size was 685.54 Mb. The long contig N50 (19.70 Mb), scaffold N50 (30.77 Mb), chromosome sizes (21.74~44.53 Mb), average transcript length (12,823.03 bp), and average CDS length (1676.66 bp), together with the high gene number (22,716), mapping ratio (97.76%), genome coverage (99.94%), and recognition rate of single-copy orthologues and core eukaryotic genes (97.2% and 93.95%), collectively suggested that the assembled male *D. maruadsi* genome was of superior quality. The genome of the male *D. maruadsi* shared a high level of collinearity or synteny with the genome of the male *T. maccoyii*, further indicating the high quality and accuracy of the assembled genome.

The genome size, contig N50, scaffold N50, chromosome number, and sizes of the male *D. maruadsi* obtained herein were comparable to those of the female *D. maruadsi* (Table 6). However, the annotation results were somewhat different because of the sample and methodological differences. For example, the average transcript and CDs lengths of the protein-coding gene of the male *D. maruadsi* genome were longer than those of the female *D. maruadsi*. In contrast, the gene number in the male *D. maruadsi* genome was slightly less than that of the female *D. maruadsi* [16]. Noteworthily, previous studies postulated that Nanopore reads have a significant 6-mer bias, whereas PacBio reads have a small 6-mer bias [61,62]. Homopolymers are difficult to be accurately called by base-callers [63]. A high deletion rate thus occurs in homopolymers in Nanopore sequencing [62].

During the evolution of species, different organisms gradually form their own unique genomes, including relatively stable DNA sequences and a fixed number of chromosomes [64]. Both male and female *D. maruadsi* genomes show that *D. maruadsi* has 23 chromosomes. The number of chromosomes in *D. maruadsi* is comparatively smaller than that of *T. maccoyii* [65], *Trachurus trachurus* [66], *S. dumerili* [67], *Trachinotus ovatus* [68], and *Seriola aureovittata* [69] (24 chromosomes) but is the same as that found in *Caranx crysos* [70], *Selene setapinnis* [71], *Gymnocypris przewalskii* [72], and *Gymnocypris eckloni* [73]. Chromosome 14 of the male *D. maruadsi* corresponded well to chromosomes 7 and 24 of the male *T. maccoyii*, while chromosome 2 of the female *D. maruadsi* (corresponding to chromosome 14 of the male *D. maruadsi*) aligned with both chromosome 2 and chromosome 4 of *T. trachurus* and *O. latipes* [16], confirming the accuracy of the chromosome number in *D. maruadsi*. The chromosome systems of fish are complex and diverse. Closely related species or even different populations of the same species may possess different chromosome systems [74]. Chromosome diversification may represent a driver of speciation and lineage diversification [75]. Studies on humans and Drosophila showed that chromosome fusion or fission could cause decreases or increases in basic chromosome numbers, which might lead to species’ reproductive isolation and promote the formation of new species [76,77,78]. Collectively, chromosome fusion might be the predominant cause of chromosome number discrepancy between *D. maruadsi* and other fish species.

Compared with other fish, the size of the chromosome-level genome obtained in this study is close to that of *S. aureovittata* (649.86 Mb) [69], *S. lalandi* (648.34 Mb) [79], *S. dumerili* (678 Mb) [67], *C. melampygus* (711 Mb) [80], and *T. ovatus* (647.5 Mb) [68], which belong to the same taxonomic family. However, the size of the chromosome-level genome obtained in this study is slightly smaller than that of *T. trachurus* (801 Mb) [66], *T. maccoyii* (782.4 Mb) (GCF_910596095.1), and *T. albacares* (792.1 Mb) (GCF_914725855.1). Fish are the oldest and largest group of vertebrates and thus have more diverse genome sizes than any other vertebrate taxon. Some studies indicate that many factors could affect fish genome sizes, such as repeat sequence content, transposable elements, and structural variations [81]. In this study, repetitive sequences accounted for 38.39% of the whole genome, of which 11.46% and 14.14% were DNA transposons and retrotransposons, respectively. Previous studies showed that transposable elements were significant contributors to genome evolution, and the proportion of transposable elements in the genome was positively correlated with the genome size across the vertebrates [82,83]. Accordingly, transposable elements may have actively contributed to *D. maruadsi* genome sizes and could be taxonomically and evolutionarily significant.

Contig N50 and scaffold N50 are widely used metrics for assessing genome quality [84]. The contigs N50 (19.7 Mb) and scaffolds N50 (30.77 Mb) of the male *D. maruadsi* genome were significantly longer than those of the *T. ovatus* (1.80 Mb and 5.05 Mb) [68] and *S. dumerili* (0.25 Mb and 5.8 Mb) [67] genomes. However, they were comparable to those of the *T. trachurus* (6.49 Mb and 35.45 Mb) [66], *S. aureovittata* (22.21 Mb and 28.35 Mb) [69], *T. maccoyii* (26.8 Mb and 33.8 Mb) (GCF_910596095.1), and *T. albacares* genomes (36.8 Mb and 34.6 Mb) (GCF_914725855.1). In addition, the total number of protein-coding genes annotated in the male *D. maruadsi* genome (22,716) was close to that in the genomes of *S. lalandi* (20,568) [79], *S. aureovittata* (21,002) [69], and *T. ovatus* (21,365) [68] in the family Carangidae. These findings collectively suggested that the male *D. maruadsi* genome obtained in this study achieved high levels of completeness, connectivity, and accuracy. They provide important genome resources for subsequent studies on resource assessment, adaptive evolution, the resolution of economic traits, and the breeding of excellent cultivars.

### 4.2. Genes Associated with Growth, Development, and Reproduction

Cytokine-cytokine receptor interaction and the JAK-STAT signaling pathway were significantly enriched in all three positive selection groups. Some genes associated with the growth and development of *D. maruadsi* in the two pathways were screened. The genes included transforming growth factor-beta 1 (TGFB1), TGFB2, bone morphogenetic protein 2 (BMP2), BMP receptor type 2 (BMPR2), BMP3, BMP3b (also known as growth differentiation factor 10, GDF10), BMP10, BMP14 (also known as GDF5), BMP15, prolactin (PRL), prolactin receptor (PRLR), platelet-derived growth factor subunit A (PDGFA), platelet-derived growth factor subunit B (PDGFB), and leptin receptor (LEPR).

The TGF-β superfamily is a ubiquitous class of multi-effector cytokines in vertebrates, including TGF-β, BMPs, GDFs, and other subfamilies. TGF-β can interact with its receptors on the surface of target cells to mediate target-intracellular signaling by activating Smad-dependent signaling pathways to regulate multiple biological processes, such as cellular proliferation, differentiation, growth control, and skeletal formation [85]. The TGF-β subfamily thus plays important roles in the growth and development process in fish. Four members of TGF-β, TGF-β1, TGF-β2, TGF-β3, and TGF-β6 were found in fish [86]. Of note, the TGF-β1 and TGF-β2 genes were significantly enriched in our study. TGF-β1 is the most widely expressed subtype of the TGF-β subfamily and is a multi-functional regulator of cell growth and differentiation. TGF-β2 delays the differentiation of muscle cells but increases cell proliferation. TGF-β1 has previously been shown to induce the differentiation of rainbow trout (*Oncorhynchus mykiss*) cardiac fibroblasts into myofibroblasts [87], inhibit zebrafish oocyte maturation at multiple sites [88,89], and limit the production of androgen and the maturation-inducing hormone (17α,20β-dihydroxy-4-pregnen-3-one) in goldfish (*Carassius auratus*) ovaries, further influencing follicular maturation as a local regulator [90]. Previous studies postulate that the expression of TGF-β2 is dynamically regulated during muscle growth resumption in rainbow trout and satellite cell differentiation [91]. TGF-β2-null mice exhibit a profound delay of hair follicle morphogenesis, with a 50% reduced number of hair follicles [92]. These reports strongly suggest that TGF-β1 and TGF-β2 potentially play a vital regulatory role in the growth and development of the ovary, cardioid, and muscle of *D. maruadsi*.

BMPs are the earliest signaling molecules that induce bone formation and differentiation. BMPs specifically bind to BMPR on the surfaces of cell membranes and transmit signals to the R-Smad, thereby activating or inhibiting the expression of genes associated with the formation of cartilage and bone, embryonic development, neural differentiation, adipogenesis, and ovarian follicle development. Numerous BMPs, such as BMP2-7, BMP8a, and BMP9-16, have been reported in fish. Among these BMPs, BMP2, BMPR2, BMP3, BMP3b, BMP10, BMP14, and BMP15 were significantly enriched in the positive selection in this study. BMP2 and BMP14 positively regulate the growth and development of bone, cartilage, and tendon by interacting with their receptors [93]. The silencing of BMP2 and BMP14 leads to the loss of joints in *Lethenteron japonicum* [94], heart malformation and shortening of pectoral and median fins in zebrafish [95,96,97], and a characteristic malformation of the scapula in mice [98]. BMP3 is a powerful negative and positive regulator of skeletal development. The significant up-regulation of BMP3 accelerates bone growth in *Sinocyclocheilus graham* [99], and the silencing of BMP3 results in the poor development of the zebrafish’s head [100,101]. Of note, BMP3-knockout mice show an increase in bone mass [102]. BMP-3b functions predominantly in bone and cartilage development and can inhibit osteoblast differentiation by antagonizing BMP-2 and -4-mediated osteogenesis [103,104,105]. BMP-3b injected into *Xenopus* embryos triggers secondary head formation autonomously, whereas the depletion of BMP-3b caused headless *Xenopus* embryos [106]. BMP3b is upregulated immediately following a fracture and is constitutively expressed at a higher level throughout osteogenesis [107]. BMP10, a cardiac-specific growth factor, promotes cell proliferation in the myocardium and plays a key role in heart development [108,109]. Silencing BMP10 in zebrafish leads to the reduction and death of cardiomyocytes [110]. Mutations in the BMP10 gene result in a hypoplastic ventricular wall, the loss of ventricular trabeculae, and a significant decrease in heat rate during mouse embryo development [111]. BMP15, a growth factor secreted by oocytes in the ovaries, plays a crucial regulatory role in the follicular development of birds and mammals [112]. Previous studies postulate that BMP-15 prevents premature oocyte maturation in zebrafish, which helps maintain oocyte quality and subsequent ovulation and fertilization [113]. Silencing BMP15 leads to decreased ovulation and reduced fertility in mice [114]. In summary, BMP2, BMPR2, BMP14, BMP3, BMP3b, BMP10, and BMP15 in *D. maruadsi* play vital regulatory roles in the growth and development processes of various tissues, such as bones, heart, muscles, and ovaries.

PRL is a single-chain polypeptide hormone produced by the anterior pituitary [115]. It binds to the PRLR and activates signaling molecules that influence gene expression and transcription associated with growth, development, reproduction, and immunity [116]. In this study, PRL and PRLR were significantly enriched, indicating that the two genes play important roles in *D. maruadsi*. Previous studies postulate that PRL and PRLR are associated with growth in *Scophthalmus maximus* [117]. In *Hippocampus abdominalis*, the accumulation of PRL in the brood pouch reduces early embryonic mortality by regulating Na^+^/K^+^-ATPase and reducing Na^+^/K^+^ concentration [118]. Additionally, PRL enhances lymphocyte proliferation and inhibits cortisol-induced proliferation and apoptosis in rainbow trout [119]. Platelet-derived growth factor (PDGF) is an important mitogenic factor and is comprised of PDGFA, PDGFB, PDGFC, and PDGFD. These factors stimulate the division and proliferation of specific cell populations and play regulatory roles in cell differentiation and ontogeny. Herein, PDGFA and PDGFB were significantly enriched and were both significantly involved in the physiological and pathological processes in the body, such as embryonic development and tissue repair. Zhang postulates that PDGFA expression is highest in the middle development stage of the tip-tissue of Sika deer antler, enhancing the proliferation rate of antler tip cells [120]. Tallquist et al. [121] and Bjarnegård et al. [122] report that PDGFA and PDGFB knockout in mice results in embryonic lethality, cardiac enlargement, and ventricular septal defect. These reports highlight the vital regulatory roles of PRL, PRLR, PDGFA, and PDGFB in the growth, development, and reproduction of *D. maruadsi.*

LEPR was also significantly enriched in the three positive selection groups of *D. maruadsi*. LEPR specifically binds to Leptin and further activates many signaling pathways (JAK/STAT, MAPK, and PI3K, among others), thereby regulating feeding, glycolipid metabolism, growth and development, reproduction, immune response, and other physiological processes [123]. The significant up-regulation of LEPR expression during early development and ovarian maturation efficiently regulates food intake, energy reserve, and reproduction in *D. labrax* and *Cynoglossus semilaevis* [124,125]. A hypothesis that LEPR and its associated genes are correlated with the growth, development, and reproduction of *D. maruadsi* is thus put forward. All genes associated with growth, development, and reproduction explored in this study are likely involved in the evolution of typical biological features in *D. maruadsi*, such as rapid growth rate, small body size, short life cycle, and strong fecundity, among other characteristics.

## 5. Conclusions

Herein, we assembled a high-quality chromosome-level genome of male *D. maruadsi* based on Illumina, PacBio, and Hi-C technologies. The size of the assembled genome is 716.13 Mb at the contig level and 716.16 Mb at the scaffold level. Notably, 23 chromosome-level genome sequences with lengths ranging between 21.74 and 44.53 Mb were assembled, and the total size was 685.54 Mb. A total of 22,716 protein-coding genes with an average transcript and CDs length of 12823.03 bp and 1676.66 bp, respectively, were annotated. Some of the protein-coding genes, such as some members of the TGF-β superfamily, are associated with the growth and development of bones, muscles, cardiac tissues, and ovaries. These genes were found to be likely involved in the evolution of typical biological features in *D. maruadsi*, such as fast growth rate, small body size, and strong fecundity. The newly established reference genome’s high contiguity, completeness, and accuracy provides a fundamental genome resource for further genetic conservation, adaptive evolution study, and genomic selection-assisted breeding of *D. maruadsi*.

## Figures and Tables

**Figure 1 animals-14-01202-f001:**
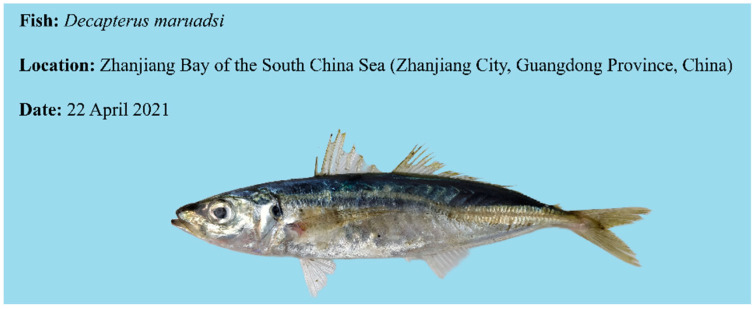
*D. maruadsi* used for sequencing.

**Figure 2 animals-14-01202-f002:**
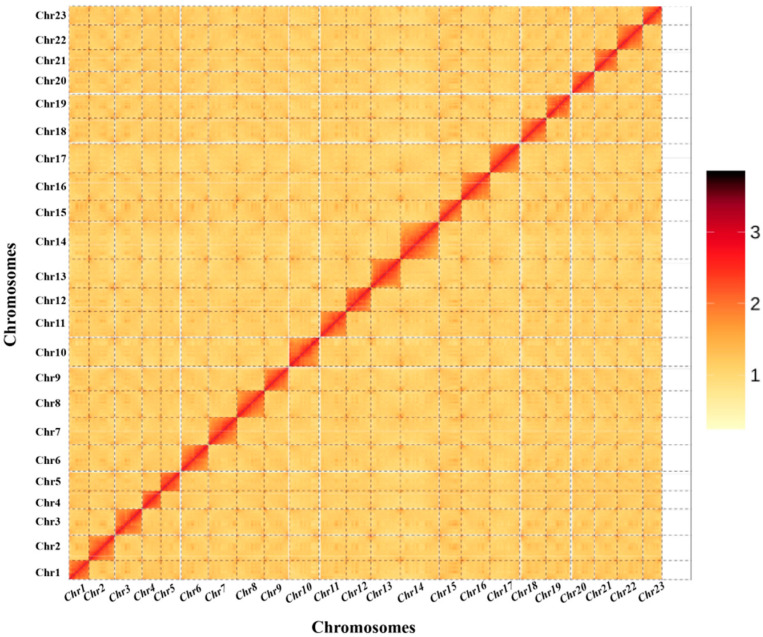
Primary chromosome contact map of the *D. maruadsi* genome based on Hi-C data. The color reflects the intensity of each contact, with deeper colors representing higher intensities.

**Figure 3 animals-14-01202-f003:**
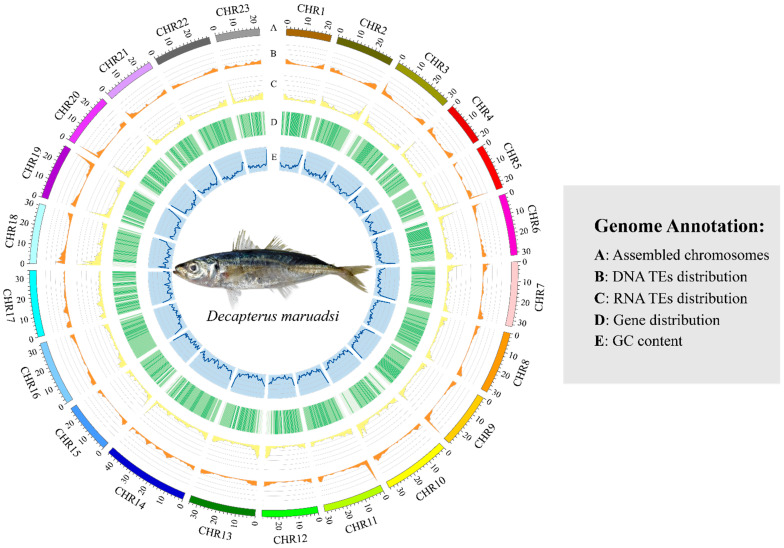
Genome coordinates and annotation information of the *D. maruadsi* genome, including (A) the length of assembled chromosomes, (B) the distribution of DNA transposable elements, (C) the distribution of RNA transposable elements, (D) the distribution of genes, and (E) the GC content of the genome.

**Figure 4 animals-14-01202-f004:**
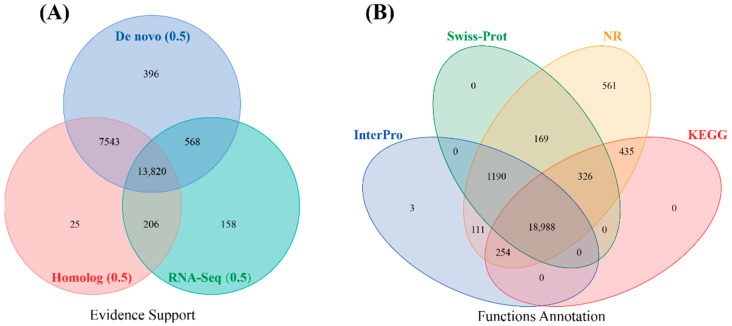
Prediction of gene structures in *D. maruadsi* (**A**). Venn diagram of functional annotation based on different databases (**B**).

**Figure 5 animals-14-01202-f005:**
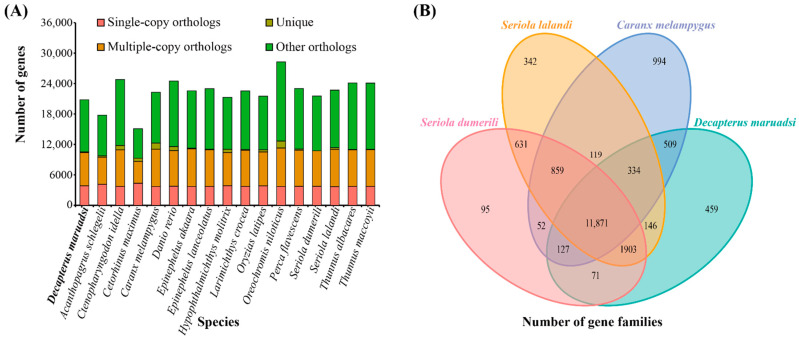
Types and numbers of gene families in 17 species (**A**) and quantitative analysis of gene families in *D. maruadsi*, *C. melampygus*, *S. lalandi*, and *S. dumerili* (**B**).

**Figure 6 animals-14-01202-f006:**
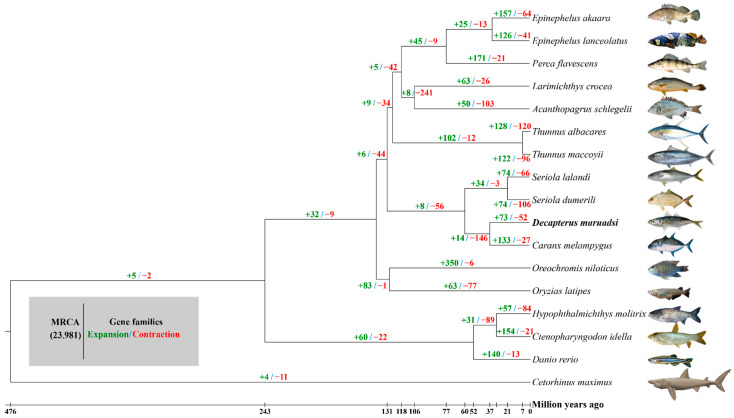
The expanded and contracted gene families of 17 fish species during the evolutionary process. MRCA: most recent common ancestor.

**Figure 7 animals-14-01202-f007:**
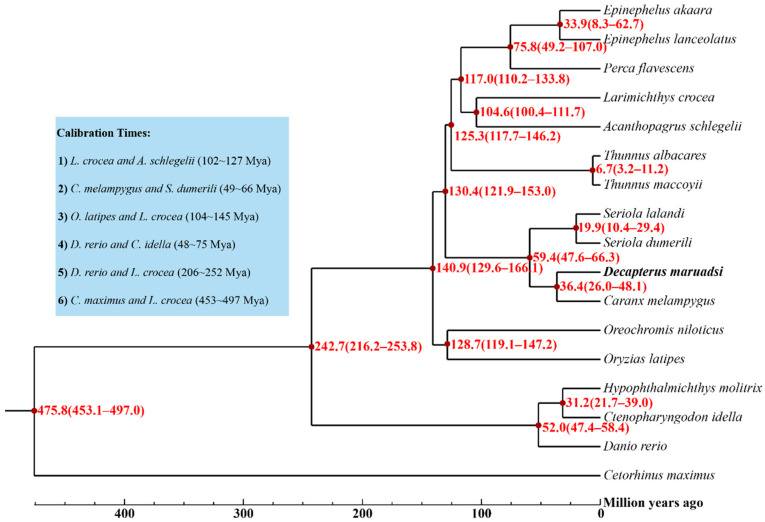
The phylogeny and divergence times of *D. maruadsi* and other fish. The red numbers on the nodes indicate the estimated divergence times. *Cetorhinus maximus* was chosen as the outgroup species.

**Figure 8 animals-14-01202-f008:**
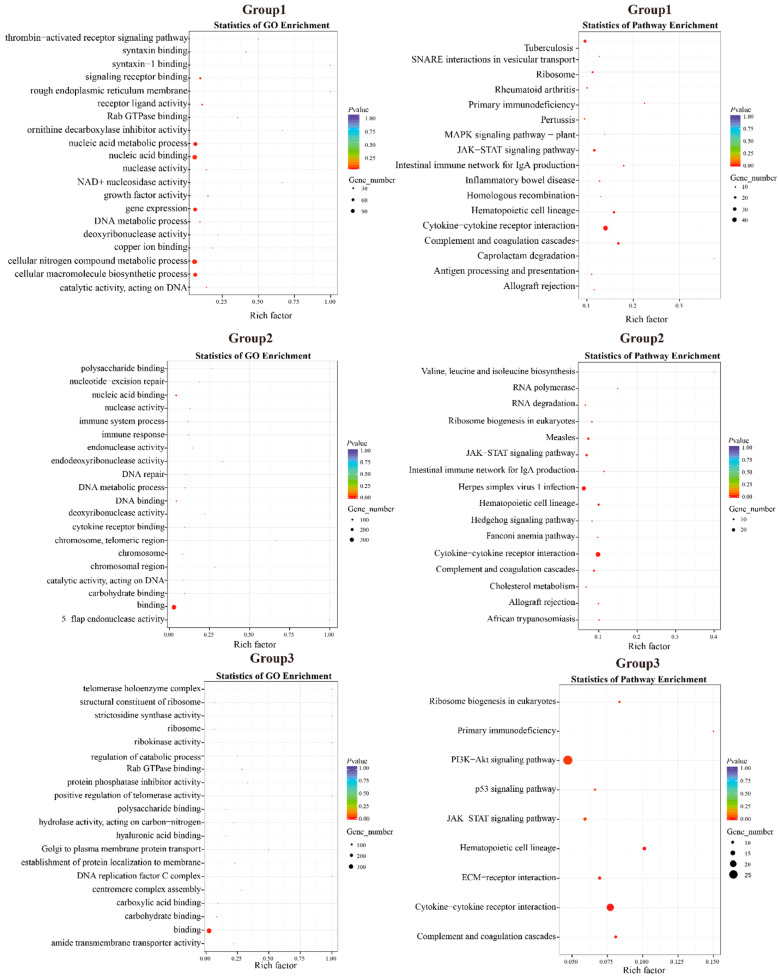
GO and KEGG enrichments of three positive selection groups. Group1: *D. maruadsi* vs. *C. melampygus*, *S. dumerili*, and *S. lalandi*. Group2: *D. maruadsi* vs. *T. albacares* and *T. maccoyii*. Group3: *D. maruadsi* vs. *L. crocea*, *E. akaara*, and *A. schlegelii*.

**Figure 9 animals-14-01202-f009:**
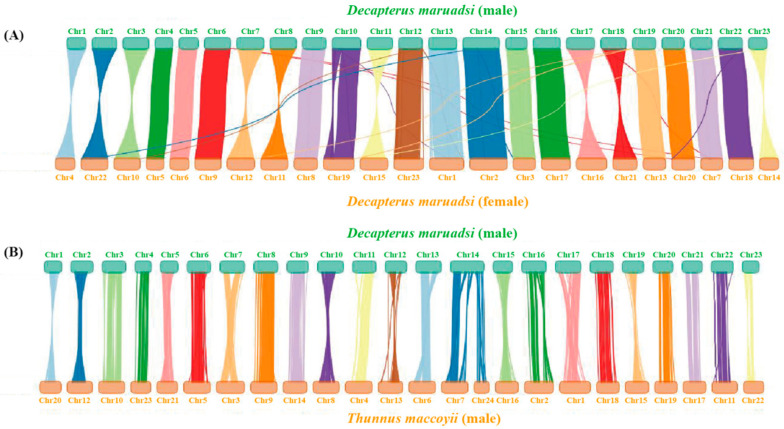
Collinearity analysis of the male *D. maruadsi* and the female *D. maruadsi* (**A**) and the male *T. maccoyii* (**B**). Each coloured line in (**A**,**B**) represents a 1 Kbp fragment match between two species.

**Table 1 animals-14-01202-t001:** Statistics of assembly data in genome sequencing.

Platform	Library Size	Raw Data	Clean Data	Coverage
Illumina DNA-Seq	350 bp	35.55 G	27.32 G	49.18×
PacBio SMRT DNA-Seq	15 kb	23.16 G	-	32.04×
Illumina Hi-Seq	350 bp	75.38 G	62.54 G	106.57×
Illumina RNA-Seq	350 bp	18.88 G	18.17 G	26.75×
PacBio RNA-Seq	-	83.33 G	81.52 G	118.06×

**Table 2 animals-14-01202-t002:** *D. maruadsi* whole-genome assembly statistics.

Sample ID	Contig Length	Scaffold Length	Contig Number	Scaffold Number
Total	716,127,322	716,155,622	357	74
Max	32,717,515	44,530,477	-	-
Number ≥ 2000	-	-	357	74
N50	19,703,568	30,768,099	14	11
N60	17,277,660	28,712,820	18	14
N70	10,918,262	28,631,201	23	16
N80	8,369,130	25,845,031	31	19
N90	1,424,252	22,449,952	57	22

**Table 3 animals-14-01202-t003:** Statistics of gene structure and functional annotation predicted by three different methods.

Methods	Gene Set	Number	Average Transcript Length (bp)	Average CDS Length (bp)	Average Exons Per Gene	Average Exons Length (bp)	Average Intron Length (bp)
De novo	Augustus	27,707	8899.69	1341.68	7.69	174.52	1130.15
	GlimmerHMM	84,170	7253.22	685.44	4.35	157.69	1962.40
	SNAP	52,373	10,543.74	840.02	5.78	145.23	2028.32
	Geneid	35,529	13,193.54	1226.37	6.17	198.82	2315.45
	GenScan	33,288	15,169.07	1517.03	8.59	176.64	1799.14
Homolog	*C. melampygus*	22,005	9440.80	1419.12	8.05	176.26	1137.62
	*D. rerio*	22,061	10,042.42	1643.80	7.92	207.44	1212.92
	*E. akaara*	25,181	10,476.56	1580.20	8.12	194.55	1249.10
	*E. lanceolatus*	23,758	11,030.34	1687.10	8.55	197.44	1238.32
	*G. aculeatus*	24,659	8854.54	1370.85	7.34	186.66	1179.61
	*H. sapiens*	18,440	10,332.37	1457.94	7.96	183.2	1275.43
	*O. latipes*	23,140	10,611.21	1735.47	8.33	208.29	1210.57
	*P. leopardus*	23,258	11,014.54	1665.17	8.51	195.76	1245.56
	*S. dumerili*	22,091	11,876.31	1678.07	9.15	183.31	1250.63
	*S. lalandi*	23,283	11,313.71	1677.08	8.81	190.27	1233.19
	*T. rubripes*	21,501	11,393.36	1723.54	8.88	194.18	1227.75
RNA-Seq	PASA	36,827	10,915.62	1482.82	8.97	165.35	1183.84
	Cufflinks	31,993	11,076.56	2484.57	8.06	308.17	1216.58
EVM (EVidenceModeler)		27,885	10,820.37	1437.70	8.2	175.25	1302.49
PASA-update *		27,384	11,297.04	1486.17	8.48	175.17	1310.91
Final set **		22,716	12,823.03	1676.66	9.65	173.83	1289.29

*: Contains UTR region; **: This final set contains the UTR region.

**Table 4 animals-14-01202-t004:** Gene family clustering in 17 fish species.

Species	Gene	Single-Copy Gene Families	Multiple-Copy Gene Families	Unique Gene Families
*D. maruadsi*	22,716	2768	12,652	45
*A. schlegelii*	18,785	2429	10,652	95
*C. idella*	32,712	3749	13,044	292
*C. maximus*	16,858	2137	9007	170
*C. melampygus*	30,852	3974	10,891	524
*D. rerio*	25,573	3477	12,254	165
*E. akaara*	23,923	3222	12,435	51
*E. lanceolatus*	23,673	3177	13,011	46
*H. molitrix*	24,571	3002	11,764	146
*L. crocea*	23,201	3101	13,021	51
*O. latipes*	21,981	2879	12,065	93
*O. niloticus*	29,430	3808	12,597	297
*P. flavescens*	23,609	3126	12,684	60
*S. dumerili*	21,740	2901	12,708	5
*S. lalandi*	24,983	3275	12,930	159
*T. albacares*	24,526	3126	13,629	23
*T. maccoyii*	24,560	3154	13,604	20
Common	23,981	2468	882	-

**Table 5 animals-14-01202-t005:** The results of the positive selection analysis of *D. maruadsi*.

**Group 1: *D. maruadsi* vs. *C. melampygus*, *S. dumerili*, and *S. lalandi*; 1233 Genes; 17 KEGG Pathways, *p* < 0.05**		
**KEGG Pathways**	***p*-Value**	**Genes Screened**
Cytokine-cytokine receptor interaction	6.74 × 10^−12^	TNFRSF9, TGFB1, MPL, LEPR, PRLR, IFNGR1, PAQR8, IL2RB, CSF2RB2, TNF, TNFRSF6B, LIFR, IL6ST, CCR9, IL10RB, IL13RA1, CCL13, BMP15, IL12A, CCL20, GDF15, CSF2RA, CD4, LEP-B, TNFRSF1B, BMPR2, GDF5, BMP3, IL12B, TNFSF15, IFNGR1L, FAS
Complement and coagulation cascades	1.10 × 10^−8^	CSMD1, F9, PRG4, SHD, PRRG4, PRRG2, F2RL2, C9, FNDC1, PLG, F5, SERPINE1, PLAUR, F3, CD55, THADA
Hematopoietic cell lineage	3.22 × 10^−7^	CSMD1, CD2, TNF, LEC, CSF2RA, CD4, CD22, CD38, CD55, CD8A
JAK-STAT signaling pathway	9.19 × 10^−5^	MPL, LEPR, PRLR, IFNGR1, IL2RB, CSF2RB2, PDGFA, LIFR, IL10RB, IL13RA1, IL12A, CSF2RA, LEP-B, CCND2, IL12B, IFNGR1L, IL22RA2
Primary immunodeficiency	0.000167663	RFXANK, CIITA, CD4, AICDA, RAG1, CD8A
Intestinal immune network for IgA production	0.000266305	TGFB1, PIGR, MAP3K14, PIGR, CCR9, AICDA
Tuberculosis	0.001444511	CSMD1, TGFB1, TIRAP, RFXANK, CABP5, CABP1, IFNGR1, CABP4, CLEC2I, CITTA, PRRT1, TNF, PVALEF, MRC1, LAMP5, IL10RB, IL12A, PLA2R1, OLR1, IL12B, CD74, IFNGR1L
Ribosome	0.002606518	RPS25, TTC4, MRPL33, RPS6, RPL12, RPS11, RPS2, RPLP1, RPS19, MRPL19, MRPS14, RPL18, MRPL13, RPS10, MRPS5
Caprolactam degradation	0.006439168	AKR1A1B, HADHA
Inflammatory bowel disease	0.006937031	TGFB1, IFNGR1, PAQR8, TNF, IL10RB, IL12A, IL12B, IFNGR1L
Rheumatoid arthritis	0.025399593	TGFB1, ATP6V1C1B, PAQR8, TNF, CTLA4, CCL13, CCL20, VEGFAA, ATP6V1E1
Antigen processing and presentation	0.02616887	RFXANK, TXNDC11, CIITA, TNF, CD4, CD8A, CD74
Homologous recombination	0.03081775	FH13, EME2, PALB2, TOP3A, BRCC3, RBBP8
SNARE interactions in vesicular transport	0.033834986	STX3, BUD23, STX8, GOSR2, STX19, VAMP8
Allograft rejection	0.035246855	TNF, IL12A, IL12B, FAS
MAPK signaling pathway—plant	0.037166119	CABP5, NME7, CABP1, CABP4, PVALEF
Pertussis	0.042868873	IRF1, CASP1, TIRAP, CABP5, PLEKHS1, CABP1, CABP4, TNF, PVALEF, IL12A, IL12B, CFL2
**Group 2: *D. maruadsi* vs. *T. albacares* and *T. maccoyii*; 810 Genes; 16 KEGG Pathways, *p* < 0.05**		
**KEGG Pathways**	***p*-Value**	**Genes Screened**
Cytokine-cytokine receptor interaction	3.01 × 10^−11^	IL20RB, FASLG, CXCL6, TNFRSF13B, IL17RC, IL6ST, TNFSF14, IL2RB, CCR6, CCL26, IL20RA, CILP2, IFNAR2, IL12A, XCL1, CSF2RA, CXCR3, CD4, CD40, IL6R, IL15RA, IL10, BMPR2, FAS
Herpes simplex virus 1 infection	0.000268892	FASLG, ZNF425, ZNF16, FAM111A, ZNF644, TNFSF14, HIC2, ALYREF, ZNF768, DAXX, ZNF436, IFNAR2, IL12A, ZNF260, TICAM2, CASP8, ZFAT, IRF9, MYNN, ZNF229, ZNF227, CD74, ZFP69, FAS
Hematopoietic cell lineage	0.000360577	CD2, LEC, CSF2RA, CD22, CD4, IL6R, CD44
RNA polymerase	0.001447104	ABBX, POLR3F, FHAB, ITPRID1, POLR1D, RPII
Complement and coagulation cascades	0.001815605	F9, PRG4, F7, C6, F3, F5, C8A, PLAUR, C5, THADA
Measles	0.002456453	FASLG, FAM111A, CD28, IL2RB, IFNAR2, IL12A, CASP8, CDKN1B, IRF9, CCND2, FAS
Intestinal immune network for IgA production	0.002919063	TNFRSF13B, CD28, PIGR, CD40, IL15RA, IL10
JAK-STAT signaling pathway	0.005214936	IL20RB, IL6ST, IL2RB, IL20RA, IFNAR2, IL12A, CSF2RA, IL6R, IL15RA, IRF9, IL10, CCND2
Valine, leucine and isoleucine biosynthesis	0.009278319	BCAT1, TD
African trypanosomiasis	0.010319507	FASLG, HMCN1, IL12A, IL10, FAS
Ribosome biogenesis in eukaryotes	0.010659323	DKC1, REXO5, RPP25, XRN1, RIOK1, UTP14A, VSTM2A
Allograft rejection	0.011171214	FASLG, CD28, IL12A, IL10, FAS
Fanconi anemia pathway	0.012071448	BRCA1, FANCM, SLX4, PALB2, ATRIP, RMI1
Hedgehog signaling pathway	0.02553919	ARR3, KIN, CFAP100, ARRB1, ZFC3H1, CCND2
Cholesterol metabolism	0.043687605	FAM43B, APOA4, TMEM259, LDLRAP1, PLTP, STAR
RNA degradation	0.049693288	EXOSC3, LSM7, PATL1, DIS3L, XRN1M WDR55, OXR1
**Group 3: *D. maruadsi* vs. *A. schlegelii*, *L. crocea*, and *E. akaara*; 761 Genes; 9 KEGG Pathways; *p* < 0.05**		
**KEGG Pathways**	***p*-Value**	**Genes Screened**
Cytokine-cytokine receptor interaction	4.70 × 10^−6^	IL20RB, IL17RB, TNFRSF13B, TUB, PRLR, IL2RB, CCR6, CCR9, TNFSF12, IL12A, CCL20, CSF2RA, IL22RA2, CD4, IL11, NGFR, BMP10, BMP3, ILFR, TNFSF15, FAS
Hematopoietic cell lineage	0.000202903	GP9, CSF2RA, CD22, CD4, IL11, CD38, KITLG, CD44, CD8A
Primary immunodeficiency	0.001047259	TNFRSF13B, CD4, RAG1, CD8A, UNG, BLNK
Complement and coagulation cascades	0.00368767	F9, PRG4, PRRG4, PRRG2, PLAU, PRG4, F3, SERPINE1, C8A, THADA
Ribosome biogenesis in eukaryotes	0.007445272	HEATR1, XRN1, UTP14A, POP1, VSTM2A, VSTM2L, REXO1
ECM-receptor interaction	0.007558111	GP9, COL9A3, PRG4, NEFH, PRG4, COL24A1, SVOP1, RELN, CD44, CCDC71
PI3K-Akt signaling pathway	0.020929921	PDGFC, COL9A3, ATLG62600, PRG4, PRLR, BRCA1, NEFH, PIK3R6, FGF3, GADD45GIP1, IL2RB, EFNA1, RAB1A, EIF4E2, COL24A1, LSR, EFNA4, NGFR, SGK1, MDM2, KITLG, RELN, CCDC71, EREG, THADA
p53 signaling pathway	0.026972984	GORAB, IGFBP5, SERPINE, CASP8, MDM2, CD82, CHEK2, FAS
JAK-STAT signaling pathway	0.030400518	IL20RB, TUB, PRLR, IL2RB, IL12A, CSF2RA, IL22RA2, IL1L, IRF9, LIFR

**Table 6 animals-14-01202-t006:** Chromosome comparison of the male *D. maruadsi*, the female *D. maruadsi*, and the male *T. maccoyii*.

Male *D. maruadsi*		Female *D. maruadsi*		Male *T. maccoyii*	
Chromosome	Length (bp)	Chromosome	Length (bp)	Chromosome	Length (bp)
Chr1	23,026,827	Chr4	23,735,607	Chr20	28,299,982
Chr2	29,845,322	Chr22	32,348,910	Chr12	33,635,709
Chr3	31,476,057	Chr10	31,747,162	Chr10	34,909,826
Chr4	21,740,779	Chr5	21,359,908	Chr23	26,533,419
Chr5	23,517,350	Chr6	22,930,063	Chr21	27,656,739
Chr6	31,525,687	Chr9	31,512,500	Chr5	35,576,159
Chr7	33,120,671	Chr12	33,226,575	Chr3	37,555,862
Chr8	31,939,948	Chr11	33,111,663	Chr9	35,071,145
Chr9	28,712,820	Chr8	28,300,000	Chr14	31,544,571
Chr10	34,370,581	Chr19	37,058,507	Chr8	35,154,190
Chr11	31,272,706	Chr15	33,635,488	Chr4	35,765,724
Chr12	28,690,070	Chr23	35,658,500	Chr13	32,533,796
Chr13	34,239,528	Chr1	41,317,494	Chr6	35,338,069
Chr14	44,530,477	Chr2	45,095,783	Chr7 Chr24	35,252,955 20,451,074
Chr15	25,845,031	Chr3	26,889,330	Chr16	30,468,816
Chr16	32,890,911	Chr17	34,731,214	Chr2	38,771,177
Chr17	34,080,461	Chr16	36,885,046	Chr1	41,002,747
Chr18	30,768,099	Chr21	29,349,286	Chr18	30,220,260
Chr19	28,631,201	Chr13	27,185,605	Chr15	31,126,207
Chr20	26,979,794	Chr20	27,911,733	Chr19	29,765,142
Chr21	26,623,482	Chr7	26,730,606	Chr17	30,337,103
Chr22	29,266,683	Chr18	29,852,700	Chr11	33,761,101
Chr23	22,449,952	Chr14	23,005,116	Chr22	26,962,447

## Data Availability

The raw sequencing reads of all libraries are available from NCBI via the accession number of SAMN40128304. The assembled genome is available in the NCBI with the accession number JBANGS000000000 via the project PRJNA1080458.

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
