# Peer review of "Whole-Genome Sequencing Analyses Reveal the Evolution Mechanisms of Typical Biological Features of Decapterus maruadsi"

_animals, 2024, doi:10.3390/ani14081202_

Round 1

Reviewer 1 Report

Comments and Suggestions for Authors

In this study, a high-quality chromosomal-level genome of male Decapterus maruadsi was assembled using Illumina, PacBio, and Hi-C technology. The genome sequencing resulted in 23 chromosome-level sequences, ranging in length from 21.74 to 44.53 Mb. Annotation of the genome identified 22,716 protein-coding genes associated with various biological processes. Positive selection analysis highlighted genes related to growth and development, shedding light on the evolutionary adaptations of D. maruadsi. This newly established reference genome serves as a crucial resource for genetic conservation, breeding programs, and exploration of adaptive evolution mechanisms.

Comments:

-       One notable aspect of the work is the high-quality genome obtained through the integration of three technologies: Illumina, PacBio, and Hi-C technology. It's important to mention that apart from the descriptive studies typically conducted when sequencing a genome of a new species, no significant additional studies have been performed. While the study paves the way for numerous new investigations, none have been deeply explored in this particular article.

-       The image quality of the fish in Figure 7 is poor. Additionally, it is necessary to verify that these images correspond to each species. For example, the image of Oryzias latipes does not seem to match the mentioned species.

-       There are too many figures in the main body of the text; some could be moved to supplementary material, leaving only the figures representing the most relevant data of the study. Many figures related to genome characterization could be included as supplementary, for example, Figure 8 and/or 10. However, I leave this to the discretion of the authors.

-       Emphasize the practical applications and potential impact of the reference genome in conservation and breeding efforts.

-       The conclusions section is overly repetitive with what has already been mentioned in the discussion. I would rewrite it differently or integrate this information into the discussion, eliminating this section altogether."

Overall, the paper effectively outlines the scope and significance of the study but could benefit from further elaboration on certain aspects for a clearer understanding by readers.

Author Response

Thanks a lot for your positive comments on the manuscript.

Reviewer 2 Report

Comments and Suggestions for Authors

This manuscript presents novel genomic data on the teleost Decapterus maruadsi.

The methods used are appropriate and well performed and the results should be considered for publication in my opinion. However, the manuscript also suffers from some issues that should be carefully addressed before it can be accepted for publication. 

Given the topic of the manuscript some additional information should be provided in the Introduction and the Discussion on available/missing cytogenetic (chromosome number, morphology etc.) and molecular data of the study species (including e.g. Hu et al. 2020 Front. Mar. Sci., 7:590564).

In the Introduction the authors report a “lack of a basic genetic study” on the species, however they also list available molecular studies (Chen et al., Li et al. and other citations) which seems to contradict the previous assumption. Please, provide a clearer evaluation of the available and missing data.

In turn, there are some sections which should be summarized (e.g. Intro lines 73-93) as they feel not essential for this type of manuscript.

Genome annotations A-E are described in a legend but not visible in Fig. 3. I also would enlarge it a bit to make it more reader-friendly. The same is true also for Figs 5A, 6, 7, 8 and 10. In all these images, species names, numeric values, enrichments etc. should be much more easy to read and understand.

Some methods should be described more in detail or referenced with relevant citations (see line-specific points below).

It also seems that some results and figures should be discussed or at least mentioned in the Discussion (see below).

The distribution of DNA transposable elements and RNA transposable elements is mentioned in Fig. 3 but not in the results or the discussion. These data could be of scientific interest and the authors should at least mention that different TEs content and distribution as well as differential amplification of heterochromatin could be taxonomically and evolutionary significant (Mezzasalma et al. 2007 Salamandra 55,:140-144; Petraccioli et al. 2007 Cytogenetic and Genome Research 157, 65-76).

In the Fig. 9 caption please specify the meaning of numeric values at nodes. I would also change “17 fish species” to something less generic.

Furthermore, Fig. 9 is not discussed at all. The authors should provide in the Discussion some description and interpretation of the obtained phylogenetic relationships and divergence time comparing their original results with already available phylogenies.

There are some unclear sentences, typos and repetitions. A spell check should be performed in different sections of the manuscript.

Some line-specific points are here below

Introduction

line 39-41: insert “order” Perciformes and delete “in the genus Decapterus”.

line 41:  species names at the start of a sentence should always be complete, not abbreviated.

line 47: the meaning of “in the same line” is unclear

line 49: I think that “resource decline” here means “population decline”. If so, the term population should be preferred, otherwise please specify what kind of resources you are referring to.

lines 73-93: this should be summarized

Methods

lines 137-142: Please, provide some details and/or relevant citations for the methods reported.

lines 165-168: some details could be provided for the use of Jellyfish and SMRTlink softwares 

lines 183-186: same for biotin incorporation, ligation and DNA purification

line 247: sequences can be “homologous”, but not species

line 250: which reference genome? Please, specify if you are mentioning here newly obtained results, citing previous studies or both

line 275: add citation for MUSCLE software.

line 276: please specify parameters, outgroups etc.  for phylogenetic analysis in RAxML

line 278: please provide web reference for TimeTree database and/or literature citation for the calibration points 

Results

line 308-309: is this GenBank?

line 308-309: “the top six species” should probably be reworded into something like “sequences with the highest coverage/identity”

line 309-312: please provide accession numbers for the sequences mentioned 

Discussion

lines 570: Please, describe these differences in the chromosome complement (e.g. number) among the mentioned species. It would also be important here to underline that chromosome diversification may represent a driver of speciation and lineage diversification (Mezzasalma et al. 2017, Zoologischer anzeiger 268, 41-46)

lines 594-598: Despite the completeness of the genome obtained, there is a quite significant difference between the identified protein-coding genes in D. maruadsi (22716) and T. trachurus (25797). How do the authors interpret and motivate this difference?

lines 610-615: references on the characteristics of TGF-β should be provided (e.g. Tzavlaki and Moustakas 2020, Biomolecules 10, 487)

lines 647-649: “X. embryos” is not a species, please check and correct.

Comments on the Quality of English Language

There are some unclear sentences, typos and repetitions. A spell check should be performed in different sections of the manuscript.

Author Response

(The authors gave the same response as above.)

Round 2

Reviewer 2 Report

Comments and Suggestions for Authors

In my opinion the authors have done a good job in revising the manuscript, which now feels improved in several of its sections.

There is just a limited number of line-specific points that should be addressed before acceptance. 

Line 173: It is better to not abbreviate SNP when first mentioned.

Line 177: It is not very clear what the authors mean with the term “separation” here. Please, specify its meaning.

Line 275: Please indicate the length of the final alignment used in the phylogenetic analysis with RAxML.

Line 277: Please, provide the accession number of the sequence used as the outgroup in the phylogenetic analysis.

Line 382-392: These lines see a repetition of lines 372-380. Either one of the two periods should be deleted.

Table 4: Species names should be in italics also in tables and references.

Reference #75: change “Mercello M.” with “Mezzasalma M.”

Reference #79: better to cite Mezzasalma et al. 2007 Salamandra 55:140-144 here, instead of Sidhom et al. 2020. In fact, the former paper deals with differential heterochromatin content in vertebrates.

Comments on the Quality of English Language

A minor spell check is still required.

Author Response

We thank the reviewer for careful reading.
